## Use of virtual consultations in an orthopaedic rehabilitation setting: how do changes in the work of being a patient influence patient preferences? A systematic review and qualitative synthesis

Anthony W Gilbert [1,2] Jeremy Jones,[2] Anju Jaggi,[1] Carl R May[3]

[1]Therapies Department, Royal National Orthopaedic Hospital, Stanmore, UK
[2]School of Health Sciences, University of Southampton, Southampton, UK
[3]Faculty of Public Health and Policy, London School of Hygiene and Tropical Medicine, London, UK

**Correspondence to**
Anthony W Gilbert;
anthony.gilbert@nhs.net

## ABSTRACT

**Objectives** To systematically review qualitative studies reporting the use of virtual consultations within an orthopaedic rehabilitation setting and to understand how its use changes the work required of patients.

**Methods** Following the Preferred Reporting Items for Systematic Reviews and Meta-Analysis statement, we conducted a systematic review of papers to answer the research question 'How do changes in the work of being a patient when using communication technology influence patient preferences?' Electronic databases were searched for studies meeting the inclusion criteria in April 2020.

**Results** The search strategy identified 2057 research articles from the database search. A review of titles and abstracts using the inclusion criteria yielded 21 articles for full-text review. Nine studies were included in the final analysis. Six studies explored real-time video conferencing and three explored telephone consultations. The use of communication technology changes the work required of patients. Such changes will impact on expectations for care, resources required of patients, the environment of receiving care and patient–clinician interactions. This adjustment of the work required of patients who access orthopaedic rehabilitation using communication technology will impact on their experience of receiving care. It is proposed that changes in the work of being a patient will influence preferences for or against the use of communication technology consultations for orthopaedic rehabilitation.

**Conclusion** We found that the use of communication technology changes the work of being a patient. The change in work required of patients can be both burdensome (it makes it harder for patients to access their care) and beneficial (it makes it easier for patients to access their care). This change will likely to influence preferences. Keeping the concept of patient work at the heart of pathway redesign is likely to be a key consideration to ensure successful implementation.

**PROSPERO registration number** CRD42018100896.

### Strength and limitations of this study

► A taxonomy of patient work will assist in understanding implementation processes.
► The use of middle-range theory has been employed to guide theorisation of the data.
► A secondary analysis of data has been employed to explain concepts which the authors had not originally intended.
► The date range of included studies (2005–2019) includes a range of technologies including the use of bespoke software, which may present different challenges to modern off the shelf software.

## INTRODUCTION
### Background
The National Health Service (NHS) Long Term Plan,[1] The UK's health service's plan to 'make the NHS fit for the future of patients', advocates digital-first primary care and envisions the use of e-consultations to become a new option for every patient. Virtual consultations can support the management of patients with long-term conditions such as musculoskeletal disease[2] where long-term management may require repeat visits for appointments with healthcare practitioners.

There are examples of virtual consultations in practice. The PhysioDirect telephone and advice service[3] is an example that was found to be safe and resulted in equivalent outcomes to face-to-face appointments for patients with musculoskeletal disorders. The visual component offered with video conferencing software offers distinct advantages over telephone consultations.[4] Research has been conducted investigating patients using Skype, a free to access video conferencing software, to access care.[5] Patients who

received telerehabilitation for knee arthritis via Skype[6] found it to be feasible and acceptable. The Virtual Online Consultations—Advantages and Limitations (VOCAL) Study[7] found video outpatient consultations to be safe, effective and convenient in appropriate situations.

The process of implementing a new intervention (such as the introduction of virtual consultations in healthcare) has been demonstrated to be dependent on how the intervention is operationalised by its users,[8] the work people do when they implement a new intervention[9] and the mobilisation of resources over time[10] across different settings.[11] Normalisation Process Theory frames implementation processes through its focus on the things people do when they implement a new intervention in practice. One study investigated nurse call takers conducting a physical assessment of patients' over a telephone helpline.[12] The study reported nurses' interactions with patients as they instructed them over the phone to perform physical manipulations. The accomplishment of a physical examination required work from patients that differs face-to-face consultations. Burden of treatment theory[13] explains how the capacity for action interacts with the work that stems from healthcare. Burden of treatment has been demonstrated to arise when the workload demands exceeds the capacity for patients with chronic obstructive pulmonary disease (COPD) and lung cancer.[14] An understanding of the factors that contribute to a change in the work for patients using virtual consultations is an important consideration for patient experience.

Research conducted in the UK found that the majority of people say they would use video consultations to consult their general practitioner about minor ailments and ongoing condition.[15] A proportion (approximately 35%) would not use this modality. Our previous research investigated whether patients preferred face to face or virtual consultations[16]: patients with atraumatic shoulder instability were offered the choice between Skype and face-to-face follow-up rehabilitation appointments. Half of patients preferred to see their rehabilitation professional in person[16] in part due to not having access and knowing how to use the software and equipment. The use of Skype changed what patients needed to do to engage in their care in our small study and this influenced their choice on whether or not to use it.

Preferences are a set of complex factors that may include enjoyment comparisons (x to y is preferred if someone enjoys x more than they enjoy y), comparative evaluations (x to y is preferred if someone thinks x is better than y), favouring (selecting x over y because x has a particular set of characteristics) or choice ranking (x is chosen over y if and only if they are faced with a choice of x over y).[17] To get past the complexities of preferences, preferences can be defined as a 'total subjective comparative evaluation'.[18] In essence, someone will prefer x over y after consideration of the alternatives, the actions, the state of affairs and the consequences of choosing each alternative. In this paper, we are interested in understanding how patient work influences patient preferences.

## Aims of this review

This paper reviews qualitative literature on the use of communication technology for patients in an orthopaedic rehabilitation setting to understand how the work of being a patient influences preference. The purpose of this paper is to develop a taxonomy of tasks required of patients using communication technology. We then consider how factors relating to these tasks influence the comparative evaluation patients face when offered the choice of a communication technology or a face-to-face consultation for orthopaedic rehabilitation.

## METHODS

A systematic review was conducted using the Preferred Reporting Items for Systematic Reviews and Meta-Analysis approach in order to answer the research question: How do changes in the work of being a patient when using virtual consultations influence patient preferences? This review was registered on the International Prospective Register of systematic reviews.[19] The protocol for this review, which forms phase 1 of the Care in Orthopaedics, Burden of Treatment and the Effect of Communication Technology (CONNECT) Project, has previously been published.[20]

MEDLINE, AMED, CINAHL, PsychINFO and SCOPUS were searched from inception on 4 April 2020. Full search terms and the search strategy is available to view in Supplementary Material (see online supplementary material 1). Articles were screened independently by two authors (AWG and AJ) with a third author (JJ) available to discuss any discrepancies (see online supplementary material 2 figure 1 for reporting).

Studies were eligible for inclusion providing they met the criteria for inclusion shown in table 1. Relevant studies were first screened by their title, and then by their abstract. Remaining texts were then read in full with all texts retained after this point for qualitative synthesis. Risk of bias was screened using the Critical Appraisal Skills Programme (CASP) Tool for qualitative studies.[21] A discussion was held, between two authors (AWG and AJ) with a third author (JJ) available to discuss any discrepancies, to decide whether included studies were of sufficient quality to include in the review.

Full texts were uploaded to QSR NVIVO (QSR International V.12, 2018), a qualitative data analysis software. NVIVO was used to collect and organise data from the results, discussion and conclusion sections of each paper. Each sentence from the included sections were coded on a line-by-line basis. The codes were labelled using a description of the content of the respective sentence. Data analysis subsequently took three forms: first, two authors (AWG and CRM) conducted a thematic analysis of codes. This was undertaken to familiarise the authors with the content of the papers. For the second iteration of coding the following was considered: What is the work of being a patient when using virtual consultations? Codes were then organised into groups

| Table 1 Eligibility criteria of studies | |
| --- | --- |
| **Inclusion** | **Exclusion** |
| ► Full-text English language academic papers from inception to 6 April 2020.<br>► Patients with an orthopaedic/musculoskeletal problem.<br>► Studies reporting patients accessing physical assessment/ rehabilitation through the use of virtual consultations (eg, telephone, video conferencing) in an orthopaedic/ musculoskeletal setting.<br>► Qualitative studies or studies with a qualitative component that focuses on the patient viewpoint of accessing virtual consultations. | ► Conference abstracts<br>► Participants without an orthopaedic/musculoskeletal complaint<br>► Quantitative studies<br>► Studies not reporting patient viewpoints |

depicting the type of work required of patients when using virtual consultations to access healthcare. The two authors (AWG and CRM) then considered the question: How might the work of being a patient when using virtual consultations influence patient preference? The data were revisited and theoretical ideas arising from the data were discussed between AWG and CRM. From here, themes, empirical regularities in the data, were identified and characterised. Finally, themes arising from the data were mapped out in the form of a model to demonstrate how, based on the included papers, the change in the work of being a patient might influence preference for virtual consultations.

## Patient and public involvement

The CONNECT Project Patient and Public Involvement Steering Group (PPISG) has been set up to provide guidance on the conduct of the research (details available from www.theconnectproject.info). The first meeting of the PPISG was held in August 2016 prior to the submission of the research to the National Institute for Health Research in May 2017. A discussion was held about the overall research aims, which supported the identification of the research questions. The PPISG has supported the design of the overall research plan and will continue to be involved during the development and refinement of each phase prior to the completion of each study protocol. In addition, the PPISG will support the development of the lay summary outputs to be disseminated to patients and members of the public. Links to research outputs will be made available on the CONNECT Project website (www.theconnectproject.info).

## RESULTS
## Study selection

Systematic search identified 1655 references (after deduplication) of which 1634 were excluded on the basis of titles and abstracts and a further 12 excluded at full-text review. As a result, nine papers were included in the review. Of the eight papers, two originated from Australia,[6 22] two from Canada[23 24] and three two from England[16 25 26] and with one from Sweden[27] and one from the Netherlands.[28] Six studies explored real-time video conferencing[6 16 23 24 26–28] and three explored telephone

consultations.[22 24 25] Study demographics are shown in table 2. All studies were screened using the CASP Tool for qualitative studies[27] and all were deemed by the authors to be of sufficient quality, and therefore retained for analysis.

## Worked example of data analysis

Data from the nine studies were synthesised. All data were treated to the same three-step process. An exemplar is demonstrated below using data from Eriksson et al[27]:

### Data identified (initial line-by-line identification)

Inability to touch the patient meant therapists were forced to rely more on their subjective assessment of the patient, leading them to spend more time talking with and listening to patients.

### Data characterised (initial line-by-line coding)

Code assigned: Therapists were unable to use 'hands on' during assessment.

### Data theorised (consideration of the question: what is the work of being a patient when using virtual consultations?)

Patients have to present themselves in a different way during assessment via virtual consultations (VC).

Data from the papers are presented in table 3.

## Synthesis of results

### Theme 1: requirements of rehabilitation
#### The processes that change

The use of virtual consultations within the treatment pathway required additional steps for patients, such as logging in Cranen et al[28] and setting the software up.[6] Some patients valued the portability of using Skype[6] and found that they could use it across different settings[16] to fulfil the purposes of the consultation. Patients valued the opportunity to run through the processes of using Skype for the first time in the form of a 'dummy run'.[16]

#### The skills and expertise that is required

The use of virtual consultations changed the skills patients needed. Video communication required specific communication skills that included listening with close attention with no interruptions.[27] The gaze of the patients and clinicians were used to signal the start and end of conversations.[27] Patients and their families found it challenging

**Table 2** Study characteristics

| Included study | Study setting | Study purpose | Technology used | Participants |
|---|---|---|---|---|
| Harrison et al[26] | Joint teleconsultations between the patient and their GP and a hospital specialist (England) | To explore patients' experiences of joint teleconferenced consultations | ISDN2 link and off-the-shelf video conferencing software | 28 patients who were enrolled in the Virtual Outreach Randomized Trial.[48] 6 patients had a generic orthopaedic diagnosis. |
| Young et al[24] | Telephone and videophone follow-up after scoliosis surgery (Canada) | To better understand the relative effectiveness of two types of telehealth technology, telephone versus videophone, following a child's scoliosis surgery from the perspective of patients and caregivers | For the videophone group, patients were provided with a videophone (KXC-AP150, Panasonic, Japan). For the telephone group patients used an ordinary telephone line | 43 patients and their families (dyads) who had undergone scoliosis correction surgery. 21 dyads received videophone support and 22 dyads who received telephone support. |
| Eriksson et al[27] | Video-based physiotherapy at the patient's home for 2 months after a shoulder replacement (Sweden) | To describe patients' experiences of physiotherapy at home by video link after a shoulder replacement | Standard commercial video conferencing units (eg, 'Tandberg 800', 'Sony PCS-50', 'Polycom VSX 3000') | 10 Adults who had undergone a shoulder replacement. |
| Cranen et al[28] | Telerehabilitation services at a rehabilitation centre (the Netherlands) | To explore patients perceptions regarding prospective rehabilitation services and the factors that facilitate or impede patients' intentions to use these services | Home-based treatment by means of (unspecified) web cam treatments | 25 chronic pain patients from a rehabilitation centre. |
| Kairy et al[23] | Telerehabilitation between the patient at home and the physical therapist at the hospital (Canada) | To better understand the patient's experience of home telerehabilitation | Internet access and the telerehabilitation platform was installed in the patient's home as reported in Wallace et al[48] The telerehabilitation device was custom built for the study | 5 patients who had previously received in-home telerehabilitation post knee arthroplasty. Patients were selected from a pool of participants from the experimental arm of a RCT for in-home telerehabilitation.[49] |
| Pearson et al[25] | Telephone-based physiotherapy between a patient and a senior physiotherapist (England) | To describe key variables that determined patient acceptability of the PhysioDirect service and to understand how the patient experience differed from those accessing usual physiotherapy care | Telephone | 57 patients with a musculoskeletal problem. Participants were recruited from the PhysioDirect Study.[50] |
| Hinman et al[6] | Skype-mediated physiotherapy consultations between the patient at home and the physiotherapist (Australia) | To explore the experience of patients and physical therapists with Skype for exercise management of knee OA | Skype software | 12 patients with a diagnosis of knee OA. Participants were key informants from an RCT.[51] |
| Lawford et al[22] | Exercise therapy for people with knee arthritis via telephone (Australia) | To explore people's perceptions of exercise therapy delivered by physiotherapists via telephone | Telephone | 20 patients with knee OA. Participants with knee OA were recruited as key informants from an RCT.[52] |
| Gilbert et al[16] | Follow-up consultations for patients after a period of inpatient rehabilitation for atraumatic shoulder instability | To explore reasons behind acceptability of Skype follow-up consultations | Skype software | 7 patients chose a Skype consultation, 6 patients chose a face-to-face consultation. In addition, 8 clinicians were interviewee. |

OA, osteoarthritis; RCT, randomised controlled trial.

**Table 3** Factors that may affect patient preference for virtual consultations and considerations for virtual consultations

| Finding | Construct | Results from included papers: factors that contribute towards the work of being a patient when using communication technology | Considerations for virtual consultations |
|---|---|---|---|
| Preferences are shaped by the requirements of the consultation how these change the work | The processes that change | Patients were able to engage in consultation from different places.[6] Using virtual consultations required patients to arrange for additional equipment in the home.[23] They were required to log in to an account[28] and to learn how to use the communication technology.[6] | ▲ Consider the impact of changing processes on patients.<br>▲ Offer troubleshooting for logging in and how to use the equipment.<br>▲ Consider offering guidance surrounding the suitability of different locations when engaging in virtual consultations. |
| | The skills and expertise that is required | As patients moved away from physically facilitated exercises, there was the requirement to adjust,[27] overcome patient–clinician communication difficulties over video call[24 26] and phone call[25] and face an increased reliance on them to communicate information.[6] In the absence of hands-on treatment, more emphasis is placed on patients completing exercises.[6] Patients need to self-assess when they cannot be physically assessed by a therapist.[6] Patients may need to adapt to clinicians who do not have adequate communication skills or training for using virtual consultations.[22] Patients may be encouraged to self-monitor improvements more than if they were seen face to face.[25] | ▲ Brief and support patients on the changes in style of communication.<br>▲ Facilitate patients to communicate their problems through a virtual consultation.<br>▲ Facilitate self-assessment of patients in the absence of clinician's 'hands-on' care.<br>▲ Facilitate and provide guidance on self-assessment and ongoing monitoring.<br>▲ Design personalised exercise regimens that are suitable for the patient's clinical problem and their home environment. |
| Preferences are shaped by the resources that are required for patients | Logistics | Use of virtual consultations helps to avoid transportation issues,[6 16 22 23 27 28] reduces travel times[28] for both patients and carers and can increase access to services.[6] | ▲ Consider offering virtual consultations for patients who experience difficulty with travel. |
| | Time | The ease in which exercises can be integrated into home routine[16 28] and through avoidance of travel provides additional time and energy for other activities.[27] Patients valued being able to wait for their appointment in their own chosen environment rather than in the clinic.[6 25 26] | ▲ Consider conflicting demands for patients.<br>▲ Consider the impact of travel and time on patient symptoms.<br>▲ Consider the impact of patient comfort when waiting for their appointment. |

Continued

**Table 3** Continued

| Finding | Construct | Results from included papers: factors that contribute towards the work of being a patient when using communication technology | Considerations for virtual consultations |
|---|---|---|---|
| Preferences are shaped by the work required due to the changes in the environment | Setting for physical rehabilitation | Patients had to find ways to overcome a lack of space[6 28] and equipment[6 23] at home. Patients were required to integrate their rehabilitation in the home environment.[6 22] | ▲ Support patients to establish a suitable rehabilitation environment at home.<br>▲ Design treatment regimens based on the patients access to rehabilitation equipment.<br>▲ Support patients to integrate rehabilitation within the home environment. |
| | Setting for virtual consultation | At times the rehab was impaired due to technical difficulties[6] and patients felt they missed learning through fellow sufferer contact through not attending the clinic and would need to seek this elsewhere.[28] | ▲ Offer troubleshooting when faced with technical difficulties.<br>▲ Consider offering peer support groups for patients who are unable to physically attend the clinic. |
| | Hardware and software | Patients needed to be supported to access[6 23] and use the equipment[6 16 24 28] and manage to real-time troubleshoot connection problems as they arose.[6 23 24 27] | ▲ Consider offering equipment based on the patient's needs.<br>▲ Tailor support for equipment use based on patient's skill set.<br>▲ Offer troubleshooting when faced with technical difficulties. |
| Preferences are shaped by the work that goes into maintaining adequate interactions | Interactions | Patients may have to focus additional attention when communicating over a stutter connection[6 27] or when faced with a language barrier.[24] Patients may need to rely on additional non-verbal communication when communicating over a screen.[26] Patients who feel alienated[28] or detached[25 28] or expect hands-on care[6 24 25] may need to invest additional effort in developing an effective therapeutic the patient clinician relationship. | ▲ Clearly communicate when the connection is impaired; be prepared to abandon and reboot the virtual consultation as required.<br>▲ Be prepared to emphasise the use of non-verbal communication.<br>▲ Have an awareness of patient preferences; patients who prefer face-to-face care may require additional input to develop a therapeutic relationship. |

to express how they felt from a distance and were reliant on the visual capabilities of the technology.[24] The lack of visual information was a concern for patients in the PhysioDirect service[25] who did not have visual cues and physical contact. The lack of physical contact meant that therapists were more reliant on information shared by patients rather than those derived from physical tasks.[6] Therapist focused on more effortful treatments such as exercises and self-management rather than providing them with hands-on care.[6] Traditional face-to-face interaction is well established and accepted. It was recognised that virtual communication required different skills and therapists' training needs, to ensure effective communication with patients, were considered in one study of telephone consultations.[22] Traditional physiotherapy patient assessment (such as 'hands-on' palpation of a joint) is not possible via Skype. As a result of this, patients were taught to self-palpate under guidance[6] and instructed how to demonstrate their range of movement over the screen. It is self-evident that visual assessment was not possible over telephone[22 24 25] and this required good communication from both therapists and patients to describe the movements. Patients felt they did not need 'hands-on' care when they were seen by an experienced therapist[27] and clinicians were more likely to encourage self-management and exercises when they were seen virtually.[25]

### Theme 2: resources
#### Logistics
Patients who underwent virtual consultations experienced reduced travel times and transportation issues[6 22 23 27 28] and was often seen as more convenient for patients, particularly those who suffered from chronic pain.[6] Virtual consultations enabled patients to access health services more easily.[23 26] Problems did arise with the PhysioDirect service where patients were unable to get through requiring them having to make multiple calls to speak to a therapist.

#### Time
Virtual consultations offered flexibility[22]: 'If I know I'm stuck at work and I can't get to see someone (the telephone) would be a good option…I can ring someone or have an appointment on the phone, and be at work doing what I need to do, and still have my appointment.' It was particularly useful for patients who had multiple commitments: 'Because life's so busy in general too, so to be able to speak to somebody in your home and then you can go on with your, you know, your next thing, is just wonderful…it just opens another brilliant option for people' as it provided more time for other activities and to integrate rehabilitation into daily life.[27]

### Theme 3: environment
#### Setting for rehabilitation
Rehabilitation in the home was welcomed by some patients as it gave them the opportunity to rehab within their own environment whereas other patients preferred

to keep their home environment separate from the clinical environment.[28] Patients found that they had a lack of space at home compared with the clinic[6 28] and could not access clinic-based equipment.[6 23] Rehabilitation required patients to troubleshoot ways to integrate their rehabilitation tasks within the home.[6 22]

#### Setting for virtual consultation
Some patients valued fellow sufferer contact and felt that through not physically attending the clinic they missed out on stimuli which kept them motivated. Rehabilitation was impaired when there were issues with connectivity and audio-visual interference disrupted the flow of the consultation.[6] Some patients felt that telerehabilitation was as good as real life and did not affect the flow of the consultation.[27]

#### Hardware and software
Patients who did not have access to equipment for virtual consultation needed to be provided with the required hardware.[23 24 27] In some cases, significant support was required for patients to understand how to use the equipment[6 24 28] and to troubleshoot connection problems when they arose.[6 23 24 27] Overcoming these barriers was an important factor in maintaining the quality of the virtual consultation and is likely to require technical support provided by the clinical team.[6]

### Theme 4: interactions
Some patients reported being more relaxed in their own home.[6] One patient, however, felt uncertain about having someone looking into their home and aborted the video consultation.[27] Virtual interactions were impaired at times there was a poor connection[6 27] or a language barrier.[24] These situations demanded additional focus and non-verbal communication[26] from the patient. The therapeutic relationship between patients and clinicians is negatively affected when patients feel alienated[28] or detached[25 28] from their clinician. Patients with an expectation of hands-on care[6 16 24 25] found virtual rehabilitation more challenging and may need to invest additional effort to maintain an effective relationship with their therapist.

## DISCUSSION
This review synthesised nine qualitative studies reporting the use of virtual consultations in an orthopaedic setting. We explored how the use of these technologies impacts on the work of being a patient. All studies in this review demonstrated that adjustments are required of patients to operationalise communication technology for virtually mediated clinical interactions. The adjustments (in the work) that a patient needs to make will have an effect on their experiences of receiving care. These experiences, whether previously lived or anticipated in the future, are likely to influence whether or not an individual finds the use of virtual consultations acceptable. The patient preference for a virtual consultation will depend on

individual circumstances. Some of these factors which might influence their decision have been and presented in a conceptual model. The model attempts to demonstrate the relationship between patient work and preference when using communication technology. The model suggests that the use of virtual consultations changes the work of being a patient. The consequences (both positive and negative) of these alterations in work may impact on the patient's experience of receiving their healthcare, their burden of treatment and their ability to engage with their healthcare. This is an important consideration for clinician, managers and policymakers.

Clinicians have to pay more attention to the patient as a result of communicating using technology compared with face-to-face consultations.[9] This appeared to be at odds with traditional consultations where physiotherapists spoke for half of the allotted time compared with patients who spoke for only 33.1%[29] in initial encounters. A study found, during a follow-up session between physiotherapists and patients, that physiotherapists spent twice as much time talking as the patients did and they relied on the use of their hands during the session.[30] In addition to the content within sessions, the relationship experienced between the clinician and the patient may differ during a virtual consultation due some patients being more relaxed at home.[9]

Some patients expected 'hands-on' treatment. The transfer of clinician manual therapy towards patient self-palpation[6] and exercise[25] may go against what is expected of therapists. The normative expectations of the patients change as a result of the geographical separation (and physical resources that can be mobilised) between patient and therapist.[31] This places particular emphasis on self-management which shifts the responsibility for health away from the state and onto the individual.[32] This is an important consideration as virtual consultations becomes increasingly used in clinical practice. The additional responsibility of self-management,[33] the change in work and tasks required to operationalise communication technology may further burden patients as they are rehabilitated virtually.

Patient viewpoints are important. Kaambwa et al[34] found in their study of older people that patients had strong preference for telehealth services that targeted individuals living in remote regions without easy access to clinic. Our previous research[16] demonstrated that distance to travel to a hospital was not the sole reason leading to the acceptability of Skype consultations and that preference is multifactorial. We found that having rehabilitation in the patient's own environment was preferred by some although bringing the clinical space into the patient's home can change the meaning of their home for them.[35] Greenhalgh et al[36] consider, among other things, what is expected of the patient when using new technologies and explains that complex tasks are more likely to lead to non-adoption.

Greenhalgh et al's VOCAL Study[7] found that the situations where patients were appropriate for video outpatient consultations only formed a fraction of the overall workload. Such situations included when close physical examination was not required and when both parties were technically confident and competent. The use of virtual consultations in these situations may increase patient work, and therefore contribute towards their burden of treatment. Patients may, therefore, opt to choose a face-to-face consultation. Sav et al[37] call for collaborative discussions to help alleviate treatment burden.

Digitally enabled services are a key focus for the UK's National Health Service over the next 10 years.[1] The use of digitally enabled services such as virtual consultations may be useful for some but add to the burden of treatment to others. Tools have been developed to assess burden of treatment.[38–42] Further research investigating the utility of tools such as these may highlight areas where digitally enabled services negatively (or positively) impact on patient experience. The work required and subsequent treatment burden for patients will differ on an individual case-by-case basis. Table 3 outlines some considerations for clinicians and policymakers considering the use of virtual consultations based on our findings from this systematic review. Further research investigating patient preference will help researchers and clinicians tailor services in a way that suits the need of patients.

Online supplementary material figure 2 in the supplementary material demonstrates how the themes from this review interact with patient preferences. The work required of a patient will influence their expectations of whether or not the use of virtual consultations is acceptable. The logistics and time required of a patient will shape the resources the patient has to dedicate towards their care. The space available and the equipment the patient has access to determine the suitability of the environment. These, coupled with the impact on patient–clinician interactions will determine patient preference for or against virtual consultations. This leads us to our first preposition: Proposition 1: The work required of patients when using virtual consultations will influence their preferences for their use.

Face-to-face consultations and communication technology consultations have different requirements. On choosing a face-to-face consultation, the patient follows the standard pathway. Choosing a communication technology consultation changes what is needed of patients. The change of work demands different skills, processes, expertise, logistical and environmental considerations. This in turn impacts on the nature of the interactions between the patient and their therapist. This leads us to our second preposition: Proposition 2: The preferences regarding the use of virtual consultations will influence the work of being a patient.

The outbreak of COVID-19 was first reported in Wuhan, China, and reached the UK on 31 January 2020. The COVID-19 virus spreads primarily through droplets of saliva or discharge from the nose when an infected person coughs or sneezes. Social distancing measures have been established with the UK public being placed on

'lockdown' from 23 March 2020[43] to avoid transmission of the disease. Healthcare organisations have subsequently embraced the use of virtual consultations to comply with these social distancing measures.[44] The outbreak of COVID-19 has led to a huge upsurge in the interest and importance of virtual consultations in practice.[44–46] As such, many more patients have been forced into undergoing virtual consultations than would have otherwise been required. A joint unit bringing together the Department of Health and Social Care, NHS England and NHS Improvement (NHSX) recently published information governance advice for health and care professionals[47] to facilitate appropriate use of virtual consultations during COVID-19. Future research should carefully evaluate the consequences of rapid virtual consultation implementation to allow for appropriate redesign of services embracing communication technology. Such redesign should consider how the use of these technologies impact on the work of being a patient.

## Limitations of this review

Our review is subjected to a number of important limitations. We included papers from the UK, Sweden, the USA, Canada, the Netherlands and Australia which used a variety of communication technologies. The data that underpin our results are a secondary analysis of other previously collected data. We did not have access to the original qualitative datasets, only that presented in the research papers. To arrive at our conclusions, we have subjected the data from the primary studies to explanatory concepts that the original authors had not intended. The studies spread from 2005 to 2018. During this time, technology has advanced considerably and the bespoke software used in the earlier studies (that were developed for the research study) may present different challenges to modern off the shelf software for use with commonly used personal devices such as phones, tablets or computers. It is also important to acknowledge the differences between the different types of technologies. A phone call does not allow for visualisation, whereas a video call does. Focusing on specific technologies may have generated more applicable results. The original research recruited patients who had opted into these studies. Patients who are satisfied with these technologies are more likely to be recruited to telemedicine studies and may not be a representative sample.

## CONCLUSION

We reviewed eight qualitative studies that reported the use of phone or video call in orthopaedic care and found that the use of virtual consultations changes the work of being a patient. We identified four different kinds of work relating to: (1) the consultation, (2) the use of resources, (3) changes in the environment and (4) interactions with the healthcare professional. Across all four domains, the change in work required of patients can be both burdensome (it makes it harder for patients to

access their care) and beneficial (it makes it easier for patients to access their care). The burden experienced by patients is a result of the relationship between the demands of the work and their capacity to fulfil these demands. Such burden is individual and situational, depending on the clinical requirements and the patient's lifeworld. As a result, we have proposed that the work of being a patient influences their preferences and the resulting choice has consequences on the resulting work that is required of them. Changes in circumstances (such as availability of equipment, understanding of how to use the equipment, requirements of the rehabilitation) may alter what is required both clinically and technologically and influence preferences. This is an important consideration to patients, clinicians, managers and policymakers, especially at a time where the use of technology is being favoured during the COVID-19 outbreak. We have demonstrated the importance of considering the work of being a patient when designing and implementing new technologies. Keeping the concept of patient work at the heart of technology implementation is essential to ensure successful uptake in practice.

**Acknowledgements** The authors thank Professor Maria Stokes, Rachel Dalton and members of the Care in Orthopaedics, Burden of Treatment and the Effect of Communication Technology (CONNECT) Project Patient and Public Involvement Steering Group for their invaluable contributions to the overall study design of the CONNECT Project and obtaining funding for the PhD Fellowship. The authors also thank Iva Hauptmannova, John Doyle and colleagues within the Therapies Directorate and Research and Innovation Centre at the Royal National Orthopaedic Hospital for their ongoing support.

**Contributors** AWG wrote the paper and conceived the project with CRM and JJ. CRM contributed knowledge on systematic reviews and qualitative analysis. AWG and AJ completed the literature search, identification of papers and quality analysis of papers. CRM, JJ and AJ edited and critically revised the paper. All authors have read and approved the manuscript. AWG is the guarantor of the manuscript.

**Funding** AWG is funded by a National Institute for Health Research, Clinical Doctoral Research Fellowship for this research project (ICA-CDRF-2017-03-025).

**Disclaimer** This paper presents independent research funded by the National Institute for Health Research (NIHR). The views expressed are those of the authors and not necessarily those of the National Health Service (NHS), the NIHR or the Department of Health and Social Care.

**Competing interests** None declared.

**Patient consent for publication** Not required.

**Provenance and peer review** Not commissioned; externally peer reviewed.

**Data availability statement** Data are available upon reasonable request.

**ORCID iD**
Anthony W Gilbert http://orcid.org/0000-0003-2526-8057

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
