## [Reviewer comments · BMJ Open]

ARTICLE DETAILS

TITLE (PROVISIONAL)	The use of virtual consultations for orthopaedics rehabilitation consultations: How do changes in the work of being a patient influence patient preferences? A systematic review and qualitative synthesis
AUTHORS	Gilbert, Anthony; Jones, Jeremy; Jaggi, Anju; May, Carl

VERSION 1 – REVIEW

REVIEWER	Lex van Velsen Roessingh Research and Development, the Netherlands
REVIEW RETURNED	21-Jan-2020

GENERAL COMMENTS	This review discusses changes in the work of being a patient when digital communication tools are introduced in orthopaedic rehabilitation. The review is conducted properly and presents a worthwhile synthesis of the results of 8 studies. The inclusion of only 8 studies on this topic does somewhat limit the generalizability and value of this study. Below, I list the major and minor issues that I have stumbled upon while reading the manuscript. Major issues: 1. In the introduction, the concept of 'work' is often mentioned (as is in the title). It remains vague however, what is meant by 'work'. The fact that it is spelled as 'work' does not help either. I would strongly urge the authors to be more clear by what they mean here. Moreover as this is one of the key concepts of the manuscript.2. There are quite some instances of sloppy writing in the manuscript (e.g., forgetting words). I have started with listing them (see minor issues), but I do not see it as the task of a reviewer to do copy editing. So please do a careful read and correct where necessary.3. For me, it was difficult to understand the difference between theme 1.1 and 1.3. The authors even mention that part of the expertise (1.3) are skills (the focus of 1.1). Could the authors somehow make the distinction between the themes more explicit? The same holds for themes 1.2 and 3.3. In both themes (the difficulty with) the use of equipment is mentioned.4. Table 3 and Figure 2 are basically the same, while Figure 2 consists of two figures (the thematic map, and the process changes). I think that the thematic map is covered in Table 3, and that the process changes add very little. Concludingly, I would remove Figure 2.
---

	5. The practical application of results is a very long section (due to the large Table) and, in my view, does not add a lot. Especially as the example scenarios are very obvious. In my view, the discussion of these scenarios is also not a practical application. I would see a practical application of these results a recommendation to managers and policy makers in rehabilitation care on how to implement a personalized approach towards the use of digital communication technologies for remote care. 6. At the moment, the propositions in the Discussion cannot be understood by themselves (see also comment #1). I would recommend the authors to rephrase these propositions to that they can be read by themselves. Minor issues:  1. Introduction: "Digital care may can support the management of patients" please correct 2. Introduction: "Communication technology, the use of digital to support" please correct, digital what? 3. The physiodirect telephone [...] resulted in equivalent physical outcomes. I am sorry, but I do not understand what physical outcomes are. 4. Limitations. When discussing the origins of the different studies, you forgot the Netherlands.
--	--

REVIEWER	Jack Pun City University of Hong Kong
REVIEW RETURNED	21-Feb-2020

GENERAL COMMENTS	Thank you for the opportunity of having me to review this interesting article. The article will create an important impact on the field of an evidence-based approach to a better understanding of technology use in health practitioner-patient communication. It answers the question "How do changes in the 'work' of being a patient when using communication technology influence patient preferences?" I enjoyed reading this manuscript. It is among the few studies synthesizing the empirical evidences on different types and modes of communication technology when communicating with patients. To strengthen the article, it will be great if the authors can review the healthcare policy in the UK if there is any regulations of technology use when communicating with patients, review the government reports on how patients' privacy issues or their personal information can be protected with different means of technology uses, highlight their findings, and perhaps their limitations, in order to create the research gap for the proposed study. Perhaps, the current version is not clear to the readers why there is the swift to more technology use, what are the motivations for healthcare providers opt for more technology use rather than traditional forms of communication (if there are still reliable and cost-effectives). Perhaps the authors can elaborate more about the patients' digital literacy has been increased, and healthcare providers can use more technology to interact with patients to deliver more health education. Limitations perhaps can be discussed explicitly on what kind of possible factors that make health providers are reluctant to adopt more technology use in their clinical practices. What are the obstacles? any proper training for the frontline health providers
---

	who are used to traditional means of communication? The authors should consider to say something about the possible limitations that are offered by more use of technology in clinical practice. Perhaps more use of technology in the clinical context, this will reduce interaction between health providers and patients. If not, why? please elaborate so that readers who are not convinced with technology use can 'buy in' this topic. It will be great if the authors can shed light on future research directions on this topic. I hope the authors would consider the above suggestions and modify the current manuscript. I am happy to review this article again.
--	---

VERSION 1 – AUTHOR RESPONSE

Reviewer 1	
1. In the introduction, the concept of 'work' is often mentioned (as is in the title). It remains vague however, what is meant by 'work'. The fact that it is spelled as 'work' does not help either. I would strongly urge the authors to be more clear by what they mean here. Moreover as this is one of the key concepts of the manuscript.	We have stopped referring to work as 'work' and have clearly defined what we mean by it in the introduction.
2. There are quite some instances of sloppy writing in the manuscript (e.g., forgetting words). I have started with listing them (see minor issues), but I do not see it as the task of a reviewer to do copy editing. So please do a careful read and correct where necessary.	Many thanks for raising this. We have addressed these within the manuscript as highlighted.
3. For me, it was difficult to understand the difference between theme 1.1 and 1.3. The authors even mention that part of the expertise (1.3) are skills (the focus of 1.1). Could the authors somehow make the distinction between the themes more explicit? The same holds for themes 1.2 and 3.3. In both themes (the difficulty with) the use of equipment is mentioned.	Themes 1.1 and 1.3 have now been combined. The theme processes (now 1.1) and hardware and software (3.3) both reference the use of the equipment with theme 1.1 focusing on the steps taken to operationalise the equipment in the context of the treatment pathway whereas theme 3.3 is the physical use of the equipment and understanding how to use it. These have been better defined.
4. Table 3 and Figure 2 are basically the same, while Figure 2 consists of two figures (the thematic map, and the process changes). I think that the thematic map is covered in Table 3, and that the process	We have placed figure 2 in the supplementary material – we hope this is acceptable.

changes add very little. Concludingly, I would remove Figure 2.	
5. The practical application of results is a very long section (due to the large Table) and, in my view, does not add a lot. Especially as the example scenarios are very obvious. In my view, the discussion of these scenarios is also not a practical application. I would see a practical application of these results a recommendation to managers and policy makers in rehabilitation care on how to implement a personalized approach towards the use of digital communication technologies for remote care.	Many thanks for this helpful comment. We have removed the table and section and rewritten this as suggested. We have included recommendations within the results table (Table 3).
6. At the moment, the propositions in the Discussion cannot be understood by themselves (see also comment #1). I would recommend the authors to rephrase these propositions to that they can be read by themselves	We have decided not to offer context independent propositions as further work is required to refine these. For example, further qualitative work carried out by us have extended these propositions. What is offered within this paper are propositions derived from the systematic review. We have rephrased these in accordance with comment #1 and we would like to thank the author for these helpful comments.
1. Introduction: "Digital care may can support the management of patients" please correct	This has been corrected.
2. Introduction: "Communication technology, the use of digital to support" please correct, digital what?	This has been rephrased.
3. The physiodirect telephone [...] resulted in equivalent physical outcomes. I am sorry, but I do not understand what physical outcomes are.	This has been rephrased.
4. Limitations. When discussing the origins of the different studies, you forgot the Netherlands.	We have included the Netherlands.
Reviewer 2	
To strengthen the article, it will be great if the authors can review the healthcare policy in the UK if there is any regulations of technology use when communicating with patients, review the government reports on how patients' privacy issues or their personal information can be protected with different means of technology uses, highlight their findings, and perhaps their limitations, in order to create the research gap for the	There aren't policies or regulations in the UK. Video-consultation systems are not classified as medical devices. There is some guidance that has recently been put out by NHSX and I have included this within the discussion section of the paper. All personal data is treated the same way, under GDPR, and after Brexit will be covered by already

proposed study. Perhaps, the current version is not clear to the readers why there is the swift to more technology use, what are the motivations for healthcare providers opt for more technology use rather than traditional forms of communication (if there are still reliable and cost-effectives). Perhaps the authors can elaborate more about the patients' digital literacy has been increased, and healthcare providers can use more technology to interact with patients to deliver more health education.	existing UK legislation (i.e. the Data Protection Act). The justification for this work has changed since the outset of COVID-19 and this has been addressed in the Introduction and Discussion section of the paper. Re: digital literacy, this is an interesting point. We have collected no data on this because the study was not designed to do so.
Limitations perhaps can be discussed explicitly on what kind of possible factors that make health providers are reluctant to adopt more technology use in their clinical practices. What are the obstacles? any proper training for the frontline health providers who are used to traditional means of communication? The authors should consider to say something about the possible limitations that are offered by more use of technology in clinical practice. Perhaps more use of technology in the clinical context, this will reduce interaction between health providers and patients. If not, why? please elaborate so that readers who are not convinced with technology use can 'buy in' this topic. It will be great if the authors can shed light on future research directions on this topic.	Covid has shown that provider organisations are not reluctant to do this when the need arises. We have included a reference to some of our other work that has been recently accepted – our whole organisation went over to remote consultation in a two-week period in March 2020. We have emphasised the limitations of using technology more in clinical practice. The paper doesn't focus on healthcare workers. This has been spelt out in the introduction within the 'aims of this review' section. The purpose of this paper is not to promote 'buy in' for technology. We have made some clearer suggestions regarding future research.

VERSION 2 – REVIEW

REVIEWER	Lex van Velsen Roessingh Research and Development
REVIEW RETURNED	01-Jun-2020
GENERAL COMMENTS	I have read the reviewers' rebuttal to the first review of the manuscript, and have re-assessed the manuscript itself. In

	general, I am happy with the way in which the authors have dealt with my comments. I have no further remarks there. I am a bit surprised, though, that the authors have re-framed the article as a COVID-19 article. The opening statement is about COVID-19, as is most of the introduction. The reviewed articles themselves, of course, do not describe situations that are affected by COVID-19. Then, in the discussion, COVID-19 is a minor point of discussion. Now, I understand that this pandemic has changed the context for eHealth enormously. But I do not think this manuscript is a COVID-19 related study (or review). I would urge the authors to go back to their original introduction. If they want to rethink their results within the light of COVID-19, the discussion section would be a perfect place.
--	--

VERSION 2 – AUTHOR RESPONSE

Comment	Response
Please state any competing interests or state 'None declared': None Declared	We have added this.
I have read the reviewers' rebuttal to the first review of the manuscript, and have re-assessed the manuscript itself. In general, I am happy with the way in which the authors have dealt with my comments. I have no further remarks there.	Many thanks for taking the time to review the paper again. We are very grateful for your constructive comments.
I am a bit surprised, though, that the authors have re-framed the article as a COVID-19 article. The opening statement is about COVID-19, as is most of the introduction. The reviewed articles themselves, of course, do not describe situations that are affected by COVID-19. Then, in the discussion, COVID-19 is a minor point of discussion. Now, I understand that this pandemic has changed the context for eHealth enormously. But I do not think this manuscript is a COVID-19 related study (or review). I would urge the authors to go back to their original introduction. If they want to rethink their results within the light of COVID-19, the discussion section would be a perfect place.	Thank you for this perspective. On reflection we agree with your point. We have done the following to the manuscript in light of these comments:  1. Removed mention of COVID-19 from the introduction. 2. Introduced COVID-19 within the discussion section.